# Serum Levels of Glutamate-Pyruvate Transaminase, Glutamate-Oxaloacetate Transaminase and Gamma-Glutamyl Transferase in 1494 Patients with Various Genotypes for the Alpha-1 Antitrypsin Gene

**DOI:** 10.3390/jcm9123923

**Published:** 2020-12-03

**Authors:** José María Hernández Pérez, Ignacio Blanco, Agustín Jesús Sánchez Medina, Laura Díaz Hernández, José Antonio Pérez Pérez

**Affiliations:** 1Pulmonology Department, University Hospital Nuestra Señora de Candelaria, Santa Cruz de Tenerife, 38010 Canary Islands, Spain; jmherper@hotmail.com; 2Spanish Registry of Alpha-1 Antitrypsin Deficiency (REDAAT), Respira Foundation, Spanish Society of Pulmonology and Thoracic Surgery (SEPAR), 08029 Barcelona, Spain; ignablanco@yahoo.es; 3University Institute of Sciences and Cybernetic Technologies, University of Las Palmas de Gran Canaria, 35018 Las Palmas, Spain; agustin.sanchez@ulpgc.es; 4Digestive System Department, University Hospital Nuestra Señora de Candelaria, Santa Cruz de Tenerife, 38010 Canary Islands, Spain; lauradiazhdez@hotmail.com; 5Institute of Tropical Diseases and Public Health of the Canary Islands, University of La Laguna, Genetic Area, 38206 Canary Islands, Spain

**Keywords:** Alpha-1 antitrypsin deficiency, liver disease, glutamate-oxaloacetate transaminase, glutamate-pyruvate transaminase, gamma-glutamyl transpeptidase

## Abstract

Background: Patients with liver disease associated with alpha-1 antitrypsin deficiency (AATD) are homozygous for the Z mutation, leading to chronic liver damage. Objective: To assess the serum levels of glutamate-oxaloacetate transaminase (GOT), glutamate-pyruvate transaminase (GPT), and gamma-glutamyl transpeptidase (GGT) in patients with different genotypes for the alpha-1 antitrypsin (AAT) gene. Methods: Patients (*n* = 1494) underwent genotyping of the *SERPINA1* gene, together with a determination of AAT and GOT and GPT and GGT transaminase levels. Patients with a deficient allele (*n* = 476) and with a normal genotype were compared. Results: A statistically significant association was found between deficient genotypes and GOT (*p* < 0.0003), GPT (*p* < 0.002), and GGT (*p* < 0.006). Comparing GOT levels in patients with *PI*Z* deficient variant versus those with normal genotype, an odds ratio (OR) of 2.72 (CI: 1.5–4.87) (*p* < 0.0005) was obtained. This finding was replicated with the *PI*Z* allele and the GPT values (OR = 2.31; CI: 1.45–3.67; *p* < 0.0003). In addition, a statistically significant association was found between liver enzymes and AAT values. Conclusion: The *PI*Z* allele seemed to be a risk factor for the development of liver damage. AAT deficient genotypes were associated with GOT, GPT, and GGT altered values. Low AAT levels were associated with high GPT and GGT levels.

## 1. Introduction

Alpha-1 antitrypsin (AAT) is a glycoprotein synthesized and secreted primarily by hepatocytes, whose main function is to neutralize excess elastase released by activated neutrophils, thereby protecting the extracellular matrix of the lungs from the harmful effects of this protease [1].

The two alleles that an individual possesses for this genetic locus are transmitted by autosomal Mendelian inheritance. The phenotypic relationship between normal and deficient alleles is partial dominance when circulating AAT levels are analyzed, or codominance when different protein variants are detected by isoelectric focusing (becoming complete dominance in the case of null alleles). Normal alleles, found in 85–90% of individuals, are called *PI*M*, while the most common deficient alleles are *PI*S* and *PI*Z*, with a prevalence among Caucasians of 3–10% and 1–3%, respectively [2].

Severe alpha-1 antitrypsin deficiency (AATD) is a hereditary condition, typically associated with *PI*ZZ* genotypes, which promotes the development of various diseases, including: chronic obstructive pulmonary disease (COPD), with onset in early adulthood in up to 50% of deficient subjects (from the age of 40, as opposed to 50–60 years for common COPD); childhood-juvenile cirrhosis in 2.5% of individuals with the *PI*ZZ* genotype; adult cirrhosis in 30% (generally men from the age of 50 years); hepatocarcinoma in 2–3% of elderly individuals with *PI*ZZ* genotype; systemic vasculitis (Wegener’s) in 2–3%; and neutrophilic panniculitis (in 1% of *PI*ZZ* individuals in the UK registry and 0.1% in the American registry) [3].

In clinical practice, most patients with AATD-associated liver disease are homozygous for the Z mutation (Glu342Lys). This genetic defect causes an abnormal folding of the PiZ protein, 80–90% of which is retained in the rough endoplasmic reticulum [4] of hepatocytes, forming highly stable polymer accumulations, which lead to a cellular stress response and chronic liver damage in some individuals [5]. The PiS protein polymerizes moderately, but its inhibitory capacity remains unchanged. However, the S and Z proteins of *PI*SZ* heterozygotes can form hepatic heteropolymers that can cause cirrhosis [6].

The transaminases glutamate-oxaloacetate transaminase (GOT), glutamate-pyruvate transaminase (GPT), and gamma-glutamyl transpeptidase (GGT) are enzymes routinely used as general laboratory markers of liver disease. GPT and GGT are expressed in hepatocytes. As well as in the liver, GOT is expressed in the myocardium, skeletal muscle, kidneys, brain, pancreas, lung, leukocytes, and erythrocytes [7].

The objective of this study was to determine whether patients with AAT deficiency have a higher risk of liver involvement according to the different genotypes.

## 2. Experimental Section

### 2.1. Study Design

An observational, cross-sectional, and descriptive study was carried out, in which a total of 1494 patients who attended the pulmonology outpatient clinic for any reason were included and analysed. They were divided into two comparable groups, those with a normal genotyping result (*Pi*MM*) and another group of subjects whose genotyping result was different *(Pi* ≠ MM)*. In addition, two comparable age-based groups were created (≤25 years and >25 years). The study was conducted in accordance with the Declaration of Helsinki. This study was approved by the ethics committee of the hospital, and all patients were informed of the study objectives and signed an informed consent. In the case of minors, their parent or guardian signed the consent.

Inclusion criteria: Patients who attended the pulmonology outpatient clinic consecutively, regardless of the reason for doing so, patients who had undergone genotyping of the SERPINA1 gene, patients whose GOT, GPT, and GGT values had been measured via blood clinical chemistry, and patients who expressed their participation in the study by signing the informed consent.

Exclusion criteria: Patients with severe chronic alcoholism or diagnosed with previous alcoholic liver disease, a history of drug-related liver toxicity, autoimmune or viral liver cirrhosis, fatty liver, viral hepatitis, haemochromatosis, or Wilson’s disease were excluded. Various analytical studies were carried out (serological determination of hepatitis virus, autoimmunity, iron levels, and copper levels). In addition, imaging studies were performed using liver ultrasound or computed axial tomography in some patients with abnormal laboratory tests to rule out involvement from other causes (neoplasms, malformations, infections, etc.).

### 2.2. Patients

Each patient underwent genotyping to check for *PI*S* alleles, *PI*Z* alleles, and rare variants of the *SERPINA1* gene. To make comparisons, the sample of patients was subdivided into two groups: 476 patients with a genetic diagnosis of AATD and 1018 with a normal genotype for the *SERPINA1* gene.

### 2.3. Genetic Analysis

The genotype was determined using so-called hybridization probes or *HybProbes* [8], which allow both real-time PCR to be performed and, after the initial amplification process, the genetic variants present in a certain region within the amplified DNA fragment to be identified. Specifically, the genotyping protocol described by Hernández-Pérez et al. was followed [9].

#### Determination of AAT and Transaminase Serum Levels

The AAT serum levels of each patient were quantified by immunonephelometry, while the serum concentrations of the enzymes GOT, GPT, and GGT were determined by standard clinical analysis procedures. Cut-off levels were those determined by the reference laboratories.

### 2.4. Statistical Analysis

The descriptive analyses of the variables were expressed as median (interquartile range (IQR)) or number (%). Differences in the distributions of patient characteristics by subgroups of outcomes were reported using differences with a 95% CI. Categorical data were compared using the χ^2^ test or Fisher’s exact test. Continuous variables were expressed as absolute (n) and variables (%). Differences between both groups were evaluated by univariate analysis and multiple logistic regression to calculate odds ratios (ORs). The different ORs are shown with their respective 95% confidence intervals (CIs). Multivariate logistic regression was used to assess independent associations. Linear correlations between clinical variables and biomarkers were evaluated by Pearson’s or Spearman’s correlation coefficient. A significant difference was considered when *p* < 0.05. Statistical analysis was performed with the IBM^®^ SPSS Statistics version 25 program.

## 3. Results

### 3.1. Baseline Characteristics

Of the 1494 patients, elevated GOT levels were observed in 5.7%, elevated GPT in 10.6%, and elevated GGT in 20.3%. Most cases were male, with a mean age of 51.4 years and a range between 1–94, a weight of 76.14 kg ranging between 36 and 152, and a median BMI of 27.67 kg/m^2^ with a range between 14.9 and 53.6. The median AAT level of the patients was 82.14 mg/dL with a range between 5 and 308.2. The rest of the baseline characteristics of the patients are shown in Table 1.

### 3.2. Relationship between Genotypes and Transaminase Levels

A statistically significant association was found between genotypes for the SERPINA1 gene and serum levels of liver enzymes, in the sense that the more deficient the genotype, the higher the enzyme elevation observed (odds ratio (OR): GOT = 25.32, *p* < 0.0003; GPT = 20.19, *p* < 0.002; GGT = 17.78, *p* < 0.006). It was found that serum GPT and GGT values changed more frequently the more deficient the genotypes (Figure 1). Similar results were observed with GOT values, excluding the Pi*ZZ genotype, although this was not so marked.

Regarding the GOT and GPT serum levels, when measuring the prevalence of exposure in patients who had a genotype with the *Pi*Z* allele and comparing them with patients of normal genotype (*Pi*MM*), an OR of 2.72 (CI: 1.5–4.87) was obtained for GOT with a significance level of *p* < 0.0005, and for GPT an OR of 2.31 (CI: 1.45–3.67) with a level of statistical significance of *p* < 0.0003 was obtained. When analyzing the GGT levels, an OR of 1.25 (CI: 0.82–1.88) was obtained, but this time without reaching statistical significance. Table 2 describes the results for genotypes containing the *Pi*Z* allele in more detail.

No statistically significant relationship was found between the *PI*S* allele and altered transaminase levels, with OR values for GOT, GPT, and GGT levels of 1.003 (CI: 0.55–1.81), 0.88 (CI: 0.56–1.37), and 0.70 (CI: 0.50–1), respectively.

### 3.3. Relationship between Transaminases and Serum AAT Levels

Finally, a statistically significant association was found between the transaminase values and the different AAT values, with a chi-squared statistical value of 14.06 for GOT (*p* < 0.002) and 17.12 for GPT (*p* < 0.0007). The result was not significant for GGT (*p* > 0.05). Our study found that low levels of AAT were associated with high levels of GPT and GGT transaminases (Figure 2).

Correlation studies showed a statistically significant (*p* < 0.001) negative correlation between AAT levels and GOT, GPT, and GGT levels.

## 4. Discussion

The liver damage associated with AAT deficiency and caused mainly by the deposition of AAT polymers is well-known, having been confirmed in the literature with various articles that all reach similar conclusions [2,10,11]. Our study shows that having the *Pi*Z* allele is a risk factor for developing liver damage, which is reflected in abnormal transaminase serum levels, especially GOT and GPT. In our results, the association of abnormal transaminase levels and the *PI*ZZ* genotype was only statistically significant with GPT, mainly due to the small number of these patients in the sample.

Although several studies cast doubt on the association of heterozygous *PI*MZ* or *PI*SZ* genotypes and liver impairment [12,13], our results suggest that heterozygous states of the *Pi*Z* allele are a risk factor for the development of abnormal serum levels of some transaminases. As such, these patients require stricter liver enzyme control, albumin levels, and coagulation status throughout their lives, and imaging tests, such as abdominal ultrasound or fibroscan, that allow for early detection of abnormalities suggestive of cirrhosis could even be considered. In addition, healthy lifestyle habits, such as physical exercise, a fat-free diet, and avoiding alcohol are advisable in patients homozygous or heterozygous for the Z allele. These findings are in line with various liver function studies and the development of cirrhosis in patients with deficient genotypes, especially with the *Pi*ZZ* genotype, showing that it is a risk factor for the development of liver cirrhosis 20 times greater than in normal individuals [14,15,16]. However, our low number of *PI*ZZ* patients is not representative to confirm the results reported in those studies. This association has also been described in *PI*MZ* and *PI*SZ* heterozygous individuals, reporting ORs that can vary between 1.8–3.1, especially in men [17,18].

Similarly, it is generally accepted that the *PI*MS* or *PI*SS* genotypes do not pose a risk of liver disease due to deposits of AAT polymers [19,20], mainly due to the fact that the polymerization of PiS polypeptides occurs in a lower percentage of molecules and more slowly than in PiZs, in which cellular inclusion bodies responsible for liver damage are not formed. The results obtained in our study show that the presence of the *PI*S* allele by itself is not a risk factor for the development of liver disease. Polymerization of mutated Z-AAT is an inhomogeneous phenomenon. In the liver, undamaged hepatocytes can coexist with hepatocytes presenting large accumulations of polymer. The cause of this difference is unknown, although it is postulated that it may be related to the different secretion capacity of AAT by the hepatocytes of the same liver [20]. Some studies have described that heteropolymers are common in heterozygous patients for the Z allele [6,21]. We believe that heterozygous patients have a risk of developing elevated transaminases in response to intrahepatic damage due to varied accumulation of polymers, although it will be necessary to assess whether this damage is evolutionary or self-limiting and does not have pathological implications.

Low levels of AAT are associated with elevated levels of GPT and GGT transaminases. Specifically, AAT values between 0–40 mg/dL were associated with the most altered GPT and GGT levels, which indicates that the most deficient genotypes and, therefore, that have lower AAT levels, are frequently associated with liver involvement, with the exception of the presence of null alleles where no liver damage has been reported.

The study by Mostafavi et al. [22] reports that GOT is the liver enzyme with the highest serum levels in *PI*ZZ* and *PI*SZ* individuals up to the age of 30 years. In the elderly, GGT is affected to a greater extent, concluding that GGT plasma levels are a more useful marker to measure liver involvement. Our study revealed similar data, probably due to the fact that most of our patients were older than 30 years. However, other authors [23] have found that in patients with the *PI*ZZ* genotype, being male and over 50 years of age, having repeated elevation of plasma transaminase levels, and having been diagnosed with diabetes mellitus or COPD were associated with the development of liver disease, so these variables are proposed as risk factors.

Patients were subsequently stratified into two groups (≤25 years and >25 years), and no statistically significant results were found for any of the three transaminases (GOT, GPT, GGT) that indicated that our patients had a higher degree of liver disease in childhood and that it progressively disappeared as adulthood was reached, although our sample was lacking a significant population of patients aged 25 or under (9.86%).

Our analysis has the limitations of a cross-sectional observational study. The temporal sequence of the variables studied could not be established, making it difficult to separate risk factors from prognostic factors. Furthermore, as no imaging study was performed using fibroscan or serial abdominal ultrasound over time, it is impossible to know which patients with altered transaminases developed liver fibrosis or cirrhosis. Despite this major limitation, our data indicate that it must be considered that patients with a *PI*Z* allele may not only have altered lung function (determined by spirometry), but also that, in these patients, liver involvement is more frequent than expected, often going unnoticed, and transaminase alteration may be the first indication of underlying liver disease. For this reason, closer long-term follow-up should be considered with serial analytical controls that include, in addition to transaminase levels, levels of bilirubin and albumin, and a complete blood count with coagulation, to detect abnormalities that guide us towards established liver damage. The influence of potential confounding factors, such as toxic habits affecting the liver, including alcoholism and drug-related liver toxicity, as well as hepatotropic viral infections, was minimized by the application of exclusion criteria.

## 5. Conclusions

In conclusion, the results of this study indicate that the presence of a *PI*Z* allele seems to be a risk factor for the development of liver involvement, since the different genotypes of AAT deficiency were associated with abnormal GOT, GPT, and GGT values. Furthermore, lower levels of AAT imply a greater involvement of GOT and GPT transaminases.

## Figures and Tables

**Figure 1 jcm-09-03923-f001:**
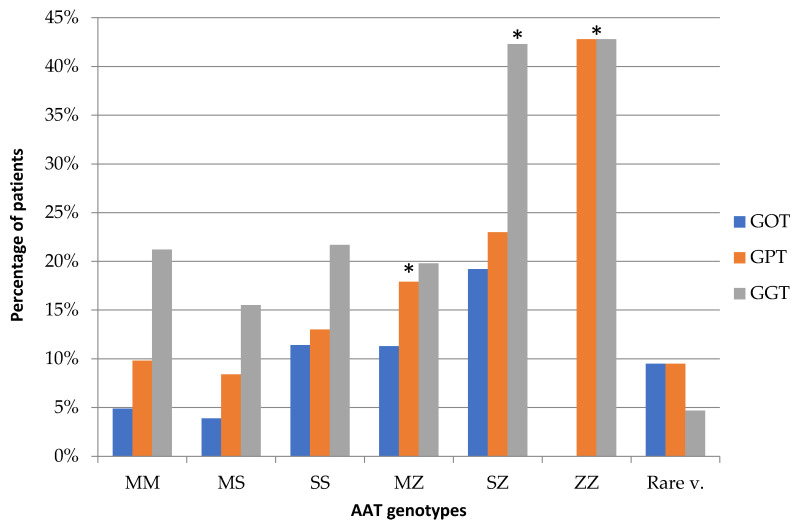
Relationship between genotypes and transaminase levels (IU/l). Transaminases (GOT > 37 U/l), (GPT > 38 U/l), and (GGT > 40 U/l). It can be seen that the most deficient genotypes are those that cause the greatest significant increases in transaminases levels (*p* < 0.05). * In the *Pi*ZZ* genotype, no changes in GOT levels were observed.

**Figure 2 jcm-09-03923-f002:**
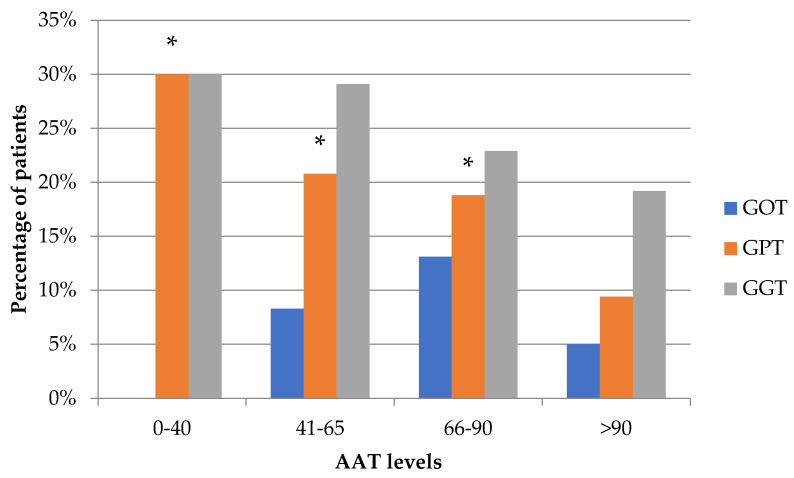
Relationship between transaminase levels (IU/L) and the different ranges of AAT levels (mg/dL). Transaminases (GOT > 37 U/L), (GPT > 38 U/L), and (GGT > 40 U/L). It can be seen that lower AAT levels were associated with statistically significant increases in transaminase levels (*p* < 0.05) *. However, in the group with the lowest AAT values (0–40), no changes were observed in GOT.

**Table 1 jcm-09-03923-t001:** Baseline characteristics of study patients. N/S: not significant. BMI: body mass index. AAT: Alpha-1 antitrypsin.

	*Pi*MM* *n* = 1018	*Pi*MS* *n* = 287	*Pi*SS* *n* = 23	*Pi*MZ* *n* = 112	*Pi*SZ* *n* = 26	*Pi*ZZ* *n* = 7	Rare Variants *n* = 21	Degree of Significance PiMM-Pi ≠ MM
Male, *n*	612 (59.18%)	142 (49.47%)	10 (43.47%)	62 (53.35%)	11 (42.30%)	4 (57.14%)	6 (28.57%)	*p* < 0.0001
Age, years, median (range)	57 (11–94)	52.05 (1–86)	57.13 (16–86)	48.68 (1–89)	46.54 (10–81)	57.86 (45–87)	40.57 (1–75)	*p* < 0.004
Weight, kg median (range)	82.22 (39–152)	79.94 (3.6–135)	74.76 (49–116)	79.67 (4.6–119)	80.52 (54–115)	65.67 (47–97)	70.22 (4.4–92)	N/S
BMI, kg/m^2^ median (range)	30.16 (14.9–53.6)	29.06 (15–52.5)	27.43 (19–40.1)	28.67 (17.9–42.2)	29.14 (21.4–44.1)	23.03 (16.3–31.3)	26.20 (19.6–37)	N/S
Serum AAT levels, mg/dL median (range)	135.9 (82.8–308.2)	115.05 (76.2–126.6)	90.31 (74.6–125)	85.15 (62.3–137.6)	60.18 (43.9–82.1)	18.94 (5–36.3)	69.51 (7–111)	*p* < 0.05

**Table 2 jcm-09-03923-t002:** Odds ratio relationship between the genotypes that include the *PI*Z* allele and the serum transaminase levels. GOT: glutamate-oxaloacetate transaminase. GPT: glutamate-pyruvate transaminase. GGT: gamma-glutamyl transpeptidase. N/S: not significant.

Genotype	GOT	GPT	GGT
*Pi*MZ*	2.49 (CI: 1.28–4.85)	4.65 (1.68–12.84)	0.91 (0.55–1.51)
	*p* < 0.005	*p* < 0.001	N/S
*Pi*SZ*	2 (CI: 1.17–3.42)	2.75 (CI: 1.07–7)	6.87 (CI: 1.52–31.15)
	*p* < 0.01	*p* < 0.02	*p* < 0.003
*Pi*ZZ*	Not applicable	2.72 (1.23–6)	2.78 (0.61–12.52)
	N/S	*p* < 0.01	N/S

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
