# Peer review of "Serum Levels of Glutamate-Pyruvate Transaminase, Glutamate-Oxaloacetate Transaminase and Gamma-Glutamyl Transferase in 1494 Patients with Various Genotypes for the Alpha-1 Antitrypsin Gene"

_jcm, 2020, doi:10.3390/jcm9123923_

Round 1
Reviewer 1 Report
The authors claim that the presence of Pi*Z allele is a risk factor for liver damage. For this they compared the Alpha1-Antitrypsin (AAT) genotype in 1494 patients with serum transaminase levels as a measure for liver problems. It is very interesting that also Pi*SZ and Pi*MZ individuals have increased liver enzymes. In addition, the authors show a relationship between lower AAT serum levels and higher transaminase serum levels.
Major comments:
1. One major limitation in this study is that the liver disease is estimated only on the basis of the liver enzymes GGT, GPT and GOT, e.g. there is no imaging performed.
2. How did the authors determine GOT, GPT and GGT cut-off levels?
3. The authors should explain why in figure 1
4. For continuous risk factors is normally the analysis of the dose response relation preferred. Why did the authors choose to make categories?
5. What did the correlation analysis give?
6. Please discuss potential confounding factors.
7. Discuss the low number of Pi*ZZ patients as a limitation of the study.
Minor comments:
- Table 1: Is it medians given for Weight, BMI and AAT levels?
- Tabel 1: If you have individuals in the age of 1 to 86 years it is hard to imagine that the weight is in the range of 36 to 135 kg. See also Pi*MZ group.
- Table 1: There must be a typo in serum AAT level range for Pi*MS.
- line 132: In the sentence "the same results were observed..." the authors should exclude Pi*ZZ genotype. Otherwise it is very irritating.
- line 194: level
- line 196: Please improve the sentence: However, ...
Author Response
Reviewer 1
- One major limitation in this study is that the liver disease is estimated only on the basis of the liver enzymes GGT, GPT and GOT, e.g. there is no imaging performed.
ANSWER: The reviewer is correct. Actually, such limitation was already stated in the text: “Furthermore, as no imaging study was performed using fibroscan or serial abdominal ultrasound over time, it is impossible to know which patients with altered transaminases developed liver fibrosis or cirrhosis”. Nevetheless, in the revised manuscript we have emphasized its nature of major limitation (line 263).
- How did the authors determine GOT, GPT and GGT cut-off levels?
ANSWER: GOT, GPT and GGT cut-off levels were those determined by the reference laboratories. This information has been added in the revised manuscript (line 128)
- The authors should explain why in figure 1
ANSWER: We think that a portion of the reviewer’s sentence is missing.
- For continuous risk factors is normally the analysis of the dose response relation preferred. Why did the authors choose to make categories?
ANSWER: We understand the reviewer’s concern. Dose response studies are typically applied to determine the association between increasing doses of an active ingredient with its local effect and intensity. However, we preferred categorization to evidence the effect of transaminases alteration in each genotype, which had not been possible if we had compared normal vs deficient genotypes instead. In the literature there are similar studies that have used the same approach, such as in references #10, #13 and #22.
- What did the correlation analysis give?
ANSWER: As the reviewer notes, in the Statistical Analysis subsection we described that the linear correlation between clinical variables and biomarkers were evaluated by Pearson's or Spearman's correlation coefficient. However, no results were provided. We apologize for the omission. In the revised version of the manuscript we have included that correlation studies showed a statisitcally significant (p<0.001) negative correlation between AAT levels and GOT, GPT, and GGT levels (line 202-203).
- Please discuss potential confounding factors.
ANSWER: Potential confounding factors included toxic habits affecting liver such as alcoholism, drug-related liver toxicity, as well as hepatotropic viral infections. In the revised manuscript potential confounding factors have been discussed (lines 169-272).
- Discuss the low number of Pi*ZZ patients as a limitation of the study.
ANSWER: The low number of PI*ZZ patients in our study is not representative to confirm the results in studies reporting that Pi*ZZ genotype is a risk factor for the development of liver cirrhosis 20 times greater than in normal individuals (refs. #14-16). This observation has been included in the revised manuscript (lines 223-224).
Minor comments:
- Table 1: Is it medians given for Weight, BMI and AAT levels?
ANSWER: Yes, it was medians. In the revised manuscritp the missing labels have been added in Table 1.
- Table 1: If you have individuals in the age of 1 to 86 years it is hard to imagine that the weight is in the range of 36 to 135 kg. See also Pi*MZ group.
ANSWER: There were some decimal points missing in weight range data of Table 1. Pi MS: 3.6-135 kg; Pi MZ: 4.6-119 kg; rare variants: 4.4-92 kg. The mistake has been ammended in the revised manuscript. We apologize.
- Table 1: There must be a typo in serum AAT level range for Pi*MS.
ANSWER: The reviewer is correct. Thecorrect range values were 76.2-126.6 mg/dL. The mistake has been ammended in the revised manuscript. Again, we apologize.
- line 132: In the sentence "the same results were observed..." the authors should exclude Pi*ZZ genotype. Otherwise it is very irritating.
ANSWER: The sentence has been reworded following the advice of the reviewer (lines 161-162).
- line 194: level
ANSWER: The missing word has been added (line 197).
- line 196: Please improve the sentence: However, ...

Reviewer 2 Report
This is an observational cross sectional study of patients attending the pulmonary clinic. patients were genotype for AAT as well as liver enzymes were obtain. the objective of the study is to show that patients with the Z allele have an increase chance of developing liver disease. several points to make.
1.- the study is not novel but the number of patients with no MM genotype as the authors describe them is significant. I think there is a value on the study but they way is written is to vague.
2.-The value of the paper lays on the data the authors describe that patients with The Z allele (carriers) do have an increase in liver enzymes when compared to patients with normal genotype.
3.- a more extended discussion on why these patients that technically should not accumulate AAT in their liver have increase in liver enzymes is warrant.
4.- the demographic data if possible should mention or exclude patients with habits or conditions that can lead to these liver changes.
5.- finally a more comprehensive plan of action these patients should be discuss. avoid etoh or other potential substance that can injure the liver. yearly screening and possible findings, ETC
Author Response
REVIEWER 2
This is an observational cross sectional study of patients attending the pulmonary clinic. patients were genotype for AAT as well as liver enzymes were obtain. The objective of the study is to show that patients with the Z allele have an increase chance of developing liver disease. several points to make.
1.- the study is not novel but the number of patients with no MM genotype as the authors describe them is significant. I think there is a value on the study but they way is written is to vague.
ANSWER: We hope that the revised manuscript is improved with the changes performed.
2.-The value of the paper lays on the data the authors describe that patients with The Z allele (carriers) do have an increase in liver enzymes when compared to patients with normal genotype.
ANSWER: Correct.
3.- a more extended discussion on why these patients that technically should not accumulate AAT in their liver have increase in liver enzymes is warrant.
ANSWER:
Polymerization of mutated Z-AAT is an inhomogeneous phenomenon. In the liver, undamaged hepatocytes can coexist with hepatocytes presenting large accumulations of polymer. The cause of this difference is unknown, although it is postulated that it may be related to the different secretion capacity of AAT by the hepatocytes of the same liver (new reference #20). Some studies have described that heteropolymers are common in heterozygous patients for the Z allele (references #6 and new #21). With all of this, we believe that heterozygous patients have a risk of developing elevated transaminases in response to intrahepatic damage due to varied accumulation of polymers, although it will be necessary to assess whether this damage is evolutionary or self-limiting and does not have pathological implications. This information has been added in the discussion of the revised manuscript (lines 232-239).
4.- the demographic data if possible should mention or exclude patients with habits or conditions that can lead to these liver changes.
ANSWER: The influence of potential confounding factors such as toxic habits affecting liver, including alcoholism and drug-related liver toxicity, as well as hepatotropic viral infections was minimized by the application of exclusion criteria. This information has been included in the discussion of the revised manuscript (lines 269-272).
5.- finally a more comprehensive plan of action these patients should be discuss. avoid etoh or other potential substance that can injure the liver. yearly screening and possible findings, ETC
ANSWER: The reviewer is correct. To prevent the development of liver disase in these patients monitoring of GOT, GPT, GGT, albumin levels, and coagulation status is recommended. Moreover, in those patients with altered transaminase levels, monitoring should also include imaging tests such as abominal ultrasound and/or fibroscan. In addition, healthy lifestyle habits such as physical exercise, a fat-free diet and avoiding alcohol are advisable in those patients homozygous or heterozygous for Z allele. This information has been completed in the Discussion of the revised manuscript (lines 218-220).

Round 2
Reviewer 2 Report
Authors have improve the manuscript.